# Genetic mechanisms control the linear scaling between related cortical primary and higher order sensory areas

Andreas Zembrzycki[1*†], Adam M Stocker[1†‡], Axel Leingärtner[1§], Setsuko Sahara[1¶], Shen-Ju Chou[1**], Valery Kalatsky[2††], Scott R May[1], Michael P Stryker[2], Dennis DM O'Leary[1]

[1]Molecular Neurobiology Laboratory, The Salk Institute for Biological Studies, La Jolla, United States; [2]Center for Integrative Neuroscience, Department of Physiology, University of California, San Francsisco, San Francisco, United States

*For correspondence:
azembrzycki@salk.edu

†These authors contributed equally to this work

Present address: ‡Biosciences Department, Minnesota State University, Moorhead, United States; §University Cancer Center Hamburg, University Medical Center, Hamburg, Germany; ¶MRC Centre for Developmental Neurobiology, Kings College, London, United Kingdom; **Institute of Cellular and Organismic Biology, Academia Sinica, Taipei, Taiwan; ††Enthought Inc, Austin, United States

Competing interests: The authors declare that no competing interests exist.

**Abstract** In mammals, the neocortical layout consists of few modality-specific primary sensory areas and a multitude of higher order ones. Abnormal layout of cortical areas may disrupt sensory function and behavior. Developmental genetic mechanisms specify primary areas, but mechanisms influencing higher order area properties are unknown. By exploiting gain-of and loss-of function mouse models of the transcription factor *Emx2*, we have generated bi-directional changes in primary visual cortex size in vivo and have used it as a model to show a novel and prominent function for genetic mechanisms regulating primary visual area size and also proportionally dictating the sizes of surrounding higher order visual areas. This finding redefines the role for intrinsic genetic mechanisms to concomitantly specify and scale primary and related higher order sensory areas in a linear fashion.

## Introduction

The mouse neocortex is patterned into functionally distinct fields that include the primary sensory areas (visual, somatosensory and auditory), which receive modality-specific sensory inputs from thalamocortical axons (TCAs) originating from nuclei of the dorsal thalamus (*O'Leary et al., 2013*). In the cortex, the connections of TCAs establish precise topographic representations (or maps) of the sensory periphery (*Krubitzer and Kaas, 2005*; *O'Leary et al., 2013*). Primary areas are flanked by higher order sensory areas (HO), which are interconnected with them and also contain topographic maps (*Felleman and Van Essen, 1991*). In mammals, this evolutionarily conserved general layout of the intra-areal neural circuits is responsible for the orderly progression of sensory information, sensory perception and the integration of higher cortical functions (*Felleman and Van Essen, 1991*; *Geschwind and Rakic, 2013*; *Krubitzer and Kaas, 2005*; *Laramée and Boire, 2014*; *O'Leary et al., 2013*). Disrupted layouts of cortical area layouts appear to be associated with neurodevelopmental disorders including autism (*Courchesne et al., 2011*; *Voineagu et al., 2011*). Studies of cortical arealization, the mechanisms that pattern the neocortex into areas, have focused almost exclusively on the primary areas and have led to the prevailing model that genetic mechanisms intrinsic to the neocortex control arealization during early cortical development (*Greig et al., 2013*; *Krubitzer and Kaas, 2005*; *O'Leary et al., 2013*). For example, the graded expression of the homeodomain transcription factor *Emx2* in neocortical progenitors determines the size and position of the primary visual area (V1) in mice (*Bishop et al., 2000*; *Hamasaki et al., 2004*). Although higher order areas outnumber primary areas by roughly 10-fold (*Marshel et al., 2011*; *Wang and Burkhalter, 2007*),

**eLife digest** The neocortex is the most recently evolved part of the human brain. It is associated with higher thought processes, including language and the processing of information from our senses. Anatomically, the neocortex is organised into different regions called 'primary areas' and 'higher order areas', and perturbations to this organisation are associated with disorders such as autism.

There are many more higher order areas than primary areas in a mammalian brain. But, while primary areas are known to be specified by developmental genes in the embryo, little is known about how the development of higher order areas is controlled. Recent findings suggested that primary areas might themselves influence the emergence of higher order areas via a series of developmental events.

Now, Zembrzycki, Stocker et al. have investigated the developmental mechanisms that organise the mouse neocortex into its different regions. The experiments involved mouse mutants that produce either too much or too little of a protein called Emx2. This protein is known to determine the size and position of the primary visual area (commonly called V1) during embryonic development. In the mutant mice with too much Emx2, the primary visual area was about 150% larger than it should be, even though the neocortex was a normal size. Zembrzycki, Stocker et al. then went on to show that the higher order areas associated with the primary visual area also grew proportionally larger in these mutant mice. The opposite was true for mice that didn't produce Emx2 in their brains, and these mice had a much smaller primary visual area than normal mice.

Together, these findings reveal a previously unknown linear relationship between the size of the primary visual area and higher order visual areas that is controlled by the genes that pattern the neocortex during development. This and other new insights will inform future studies of the development and organization of the neocortex and our understanding of how it evolved.

mechanisms that specify them and define their proportions relative to primary areas have yet to be explored.

## Results

To investigate the impact of altered primary area size on higher order areas, we have used the cortical visual area V1 as a model. Previous studies have shown that genetic manipulation of patterning genes, including *Fgf17* and *Emx2,* results in altered V1 size (*Cholfin and Rubenstein, 2007*; *Hamasaki et al., 2004*). Here we have analyzed transgenic mice that overexpress *Emx2 (ne-Emx2)* and show area patterning defects including a V1 that is ~150% of the normal size, while retaining overall normal neocortex size (*Hamasaki et al., 2004 Leingärtner et al., 2007*). By revealing the targeting patterns of TCAs projecting from thalamic sensory nuclei into the cortex (*Fujimiya et al., 1986*), the perimeters of primary sensory areas and the border between the neocortex and entorhinal cortex (ECT) can be visualized by serotonin (5HT) staining using a single postnatal day (P) 7 tangential section of the flattened cortical hemisphere (*Figure 1A*). The staining shows that, in addition to the previously reported enlarged V1 (*Hamasaki et al., 2004,*; *Leingärtner et al., 2007*), the cortical tissue that is nested between V1 and the surrounding primary areas (primary somatosensory cortex: S1, auditory areas: Aud) and the ECT laterally appears qualitatively larger in *ne-Emx2* brains, when compared to wildtype (wt) sections (*Figure 1A*). We have defined this caudal cortical territory, which lies outside of V1, S1, and the auditory areas and shows no or weak 5HT staining as a joint higher order cortical area complex and have termed it HO-5HT. The 5HTstaining revealed that this region contains the higher order visual areas surrounding V1 (*Wang and Burkhalter, 2007*), the retrosplenial cortex (RSC) medially, and the ventral posterior temporal cortex laterally. The accurate distribution of staining across cortical layers can only be estimated using tangential sections. However, using P7 sagittal section, we confirmed in layer 4 that the caudal 5HT-positive cortical area (corresponding to V1) and the anteriorly adjacent 5HT-negative area between V1 and S1 (corresponding to HO-5HT) appears larger in *ne-Emx2* brains than in wt ones (*Figure 1B*). Next, we labeled TCAs projecting to V1 by filling the dLG with crystals of the lipophilic neuronal tracer DiI. On

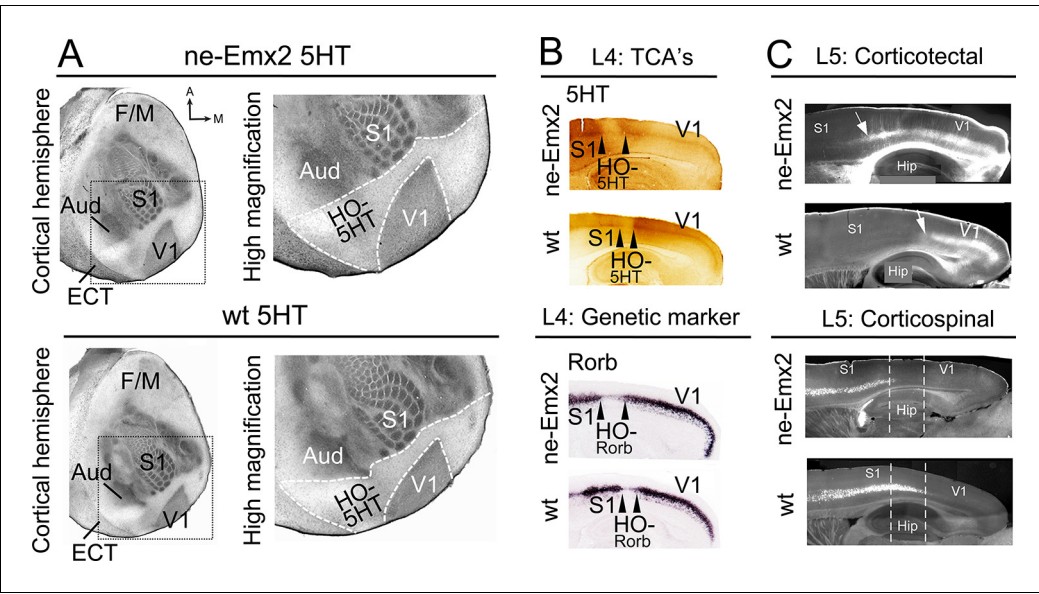

**Figure 1.** Increased V1 and higher order sensory area sizes in *ne-Emx2* cortices (**A**) Serotonin (5HT) staining on postnatal day (P) 7 tangential sections of the flattened cortex reveals targeting patterns of TCAs revealing primary sensory area borders and the border of the neocortex to the ECT. 5HT staining is not detectable in the region containing the retosplenial cortex and the higher order sensory areas surrounding V1 (HO-5HT). In *Emx2*-overexpressing brains (*ne-Emx2*), V1 and HO-5HT appear larger (compare dotted outlines in higher magnification images), compared to wt brains. (**B**) Targeting of TCAs in cortical layer 4 (L4) was revealed on P7 sagittal cortex sections by 5HT staining, whereas L4 genetic area borders were revealed by in situ hybridization for *Rorb*. In *ne-Emx2* brains, the V1 border shifts anteriorly. Higher order areas surrounding V1 are characterized by low 5HT/*Rorb* staining (between arrowheads, HO-5HT and HO-*Rorb*), which in *ne-Emx2* brains appear overall larger (compare area between arrowheads). (**C**) In L5, an expansion (see arrows) of corticotectal projection neurons (retrogradely labeled by DiI injections into the superior colliculus) is apparent in *ne-Emx2* brains, to the expense (see dotted lines) of L5 corticospinal projection neurons (retrogradely labeled by DiI injections into the pyramidal decussation). Main axes: A: anterior; M: medial; F/M: frontal/motor cortex; S1: primary somatosensory cortex; Aud: auditory areas; V1: primary visual cortex, ECT: entorhinal cortex. 5HT, serotonin; L5, cortical layer 5; TCAs, thalamocortical axons; wt, wildtype.

The following figure supplement is available for figure 1:

**Figure supplement 1.** Anterior shifted cortical boundary formed by TCAs from the dLG in *ne-Emx2* brains.

the P7 sagittal sections that were derived from five different medial to lateral levels, anterograde DiI labeling in the cortex revealed that TCAs from the dLG terminate in a smaller region in wt than in *ne-Emx2* brains (*Figure 1—figure supplement 1*). Across genotypes, the DiI staining revealed a sharp border with adjacent cortical tissues that did not receive TCAs input from the dLG (*Figure 1— figure supplement 1*). This finding is consistent with the 5HT staining and indicates a well-defined border between V1 neighboring higher order areas that is anteriorly shifted in *ne-Emx2* brains.

Cortical areas can also be distinguished by area-specific gene expression patterns, which overlap with anatomical area borders and shift similarly when area patterning is disrupted (*O'Leary et al., 2013*). For example, *Rorb* expression is strongly induced by thalamic input to primary areas (*Jabaudon et al., 2012*) like S1 and V1 but is low in areas that do not receive their major inputs from the principal thalamic sensory nuclei, such as cortical higher order areas surrounding V1 (*Chou et al., 2013*; *Wang and Burkhalter, 2007*). In situ hybridization (ISH) on sagittal sections adjacent to 5HT-stained ones revealed sharp *Rorb* gene expression borders between areas in layer 4. Notably in *ne-Emx2* brains, the high-to-low *Rorb* expression border is located more anteriorly, and the area showing low *Rorb* expression and resembling HO-5HT (*Chou et al., 2013*) is larger than in wt sections (*Figure 1B*). This reveals that characteristic molecular markers that delineate the borders between

V1 and surrounding higher order areas remain expressed at normal levels, but their sharp expression borders shift anteriorly in *ne-Emx2*.

Projection neurons in layer 5, which extend axons into subcortical targets, are similarly determined by a molecular code (*Greig et al., 2013*). We therefore predicted that the areal shifts in *ne-Emx2* brains would be accompanied by corresponding changes in layer 5 output projections. We labeled two distinct types of layer 5 subcerebral projection neurons by inserting DiI crystals either into the superior colliculus, which labels corticotectal projections from V1 and HO, or else into the pyramidal decussation, which labels corticospinal projections from the frontal cortex and S1 (*Greig et al., 2013*; *Zembrzycki et al., 2015*). We found that layer 5 corticotectal projections extended more anteriorly in *ne-Emx2* sagittal sections. Vice versa, the layer 5 corticospinal projections extended less posteriorly in *ne-Emx2* brains. These staining patterns are consistent with an altered balance of projection neuron identity in layer 5 (*Greig et al., 2013*; *Zembrzycki et al., 2015*) and an overall expansion of visual areas, demonstrating that areal patterning changes in *ne-Emx2* brains are not limited to the cortical layers that receive thalamic input. These findings complement previous reports describing *Emx2* patterning functions (*Bishop et al., 2000*; *Hamasaki et al., 2004*; *Leingärtner et al., 2003*; *Leingärtner et al., 2007*) and indicate for the first time that V1 and higher order area sizes are altered concomitantly in *ne-Emx2* brains at the level of area-specific connectivity and gene expression in multiple cortical layers. Taken together, our results suggest that changes in primary area size are paralleled by similar changes in higher order area size.

It is commonly assumed that areal patterning changes also alter area-specific functional neuronal properties and topographic sensory maps, but this has never been demonstrated conclusively. Therefore, to compare functional neuronal properties of an enlarged visual cortex to a normal-sized one, we have used Fourier intrinsic signal optical imaging to construct topographic visual response maps to light bars that were moved across the visual field of the retina (up and down: elevation maps; left to right: azimuth maps) (*Kalatsky and Stryker, 2003*). Visual responses in V1 of wt and *ne-Emx2* mice produced intrinsic signal maps that were indistinguishable in strength, and the axes of azimuth and elevation were organized in the same way in all tested brains (*Figure 2A*), revealing that functional topographic organization of the visual cortex was intact. However, the representations of elevation and azimuth were expanded in *ne-Emx2* animals, and their retinotopic maps were overall larger (elevation: 138% ± 8.7% of wt; azimuth: 143% ± 8.2% of wt). For example, the green region in the response maps representing ~20 to ~30 degrees of elevation/azimuth is clearly enlarged in *ne-Emx2* brains, compared to wt brains (*Figure 2A*). To investigate the relationship between the location and size of the V1 functional response area and the histochemically delineated V1, as indicated by 5HT staining, multiple injections of DiI were placed lining the border of the V1 optical response map after the imaging procedure. On 5HT-stained, flattened tangential cortical sections, the DiI injection sites were found in all cases to be located near the border of the 5HT staining in V1, confirming the overall enlarged V1 perimeters in *ne-Emx2* brains compared with the wt brains (*Figure 2B*). This shows that the 5HT-stained V1 area accurately corresponds to the intrinsic functional V1 map, suggesting that enlarged HO in *ne-Emx2* brains have not acquired ectopic V1-like functional properties.

To further characterize the shifted border between visual areas in *ne-Emx2* animals, we used additional neuronal tracing approaches. Stereotypically, V1 is connected with the dLG, whereas the HO areas are wired to the posterior thalamic nucleus (PO) (*Leyva-Díaz and López-Bendito, 2013*; *López-Bendito and Molnár, 2003*). To first label axonal connections between the cortex and the thalamus in wt brains, we administered dual tracer injections into locations that approximate to HO (DiD: green dye) and another injection around the approximated border area between V1 and HO (DiI: red dye). After diffusion of the tracers, we performed 5HT staining on flattened cortex sections to identify the areas in which the injections were administered and analyzed the patterns of retrograde dye labeling on coronal sections of the thalamus. In representative cases (*Figure 2C*) where 5HT staining confirmed that DiI was injected at the border between V1 and HO and DiD was injected into HO, retrogradely labeled green DiD cells were apparent in the dLG and the PO. Conversely, red DiI cells were only labeled in the dLG. In *ne-Emx2* brains, we administered similar dual tracer injections: DiI was targeted to V1 and a DiD injection was administered around the approximate border between V1 and HO. Due to their enlarged V1, all *ne-Emx2* injections were administered at more anterior coordinates than in wt brains (compare dashed lines in *Figure 2C*). In representative cases (*Figure 2C*) where 5HT staining confirmed that the DiI injection was

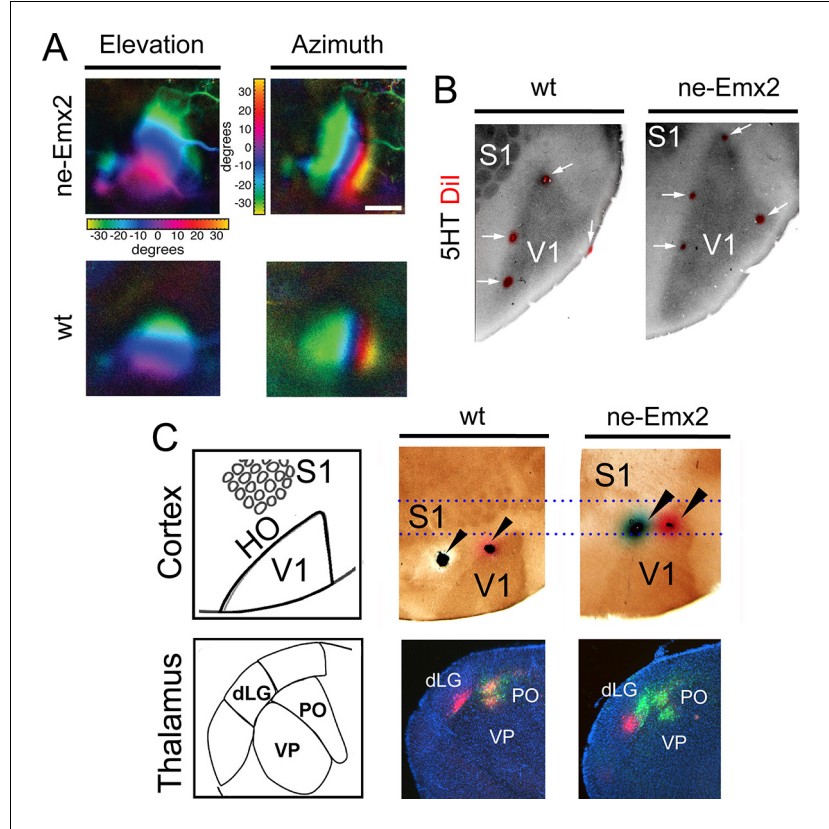

**Figure 2.** Enlarged functional V1 topographic maps in *ne-Emx2* mice. (**A**) Fourier intrinsic signal optical imaging reveals topographic visual response maps to light bars that were moved across the visual field of the retina (up and down: elevation maps; left to right: azimuth maps). Visual responses in V1 produced intrinsic signal maps that were indistinguishable in strength and the color-coded axes of azimuth and elevation were organized in the same way in all tested brains (wt: n = 6, *ne-Emx2*: n = 6). The elevation/azimuth representations were expanded in *ne-Emx2* animals, revealing that overall their V1 retinotopic maps were larger. (**B**) 5HT staining performed on flattened tangential cortex section reveals cortical area borders including V1. On representative images (n = 6 per genotype), the red dots lining the perimeter of the 5HT-stained V1 indicates DiI injection sites that were made after the recordings adjacent to the border of the derived V1 intrinsic response maps, determined by Fourier intrinsic optical imaging. (**C**) Schematics depict caudal cortical sensory areas and main sensory thalamus divisions. In wt brains (n = 15), cortical dual tracer injections (red tracer (DiI) injected around V1/HO border; green tracer (DiD) into HO; injection site location (arrowheads) was identified by 5HT staining) showed retrogradely labeled red cells in the dLG and the PO, whereas green labeled cells were only present in the PO. Dotted lines show that dual tracer injections in *ne-Emx2* brains (n = 17) were administered at more anterior coordinates (red tracer into V1; green tracer around the V1/HO border) compared with wt brains. In *ne-Emx2* brains, retrogradely labeled red cells were apparent in the dLG, whereas green cells were present in the dLG and the PO, revealing normal thalamocortical connectivity patterns, but an anterior shifted V1/HO border in *ne-Emx2* brains. 5HT, serotonin; dLG, dorsal lateral geniculate nucleus; PO, posterior thalamic nucleus; S1, primary somatosensory cortex; VP, ventroposterior nucleus; V1, primary visual cortex; wt, wildtype.

administered into V1 and DiD was injected around the V1/HO border, red cells were found in the dLG, whereas green-labeled cells were apparent in the PO and the dLG. The dual tracings in wt and *ne-Emx2* brains are consistent with the predicted connectivity of cortical neurons around the injection sites (*Leyva-Díaz and López-Bendito, 2013*; *López-Bendito and Molnár, 2003*). Although located more anteriorly in *ne-Emx2* brains, the subcortical connectivity patterns around the V1/HO borders were similar, demonstrating that these neurons show connectivity patterns that are consistent with their intrinsic areal identity and not their topographic location on the cortical sheet. Taken together, our results indicate that increased V1 size in *ne-Emx2* brains is accompanied by a concomitant enlargement and anterior shift of HO.

To define the individual magnitudes of the V1 and HO size increases in *ne-Emx2* brains, we next used gene expression domains as molecular markers delineating visual areas and quantified them (*Figure 3*). An accurate assessment of area sizes using flattened and/or sectioned cortical tissues could potentially be hampered by imperfect flattening of the tissues or by cutting artifacts. Therefore, we have used RNA in situ hybridization on intact whole brains (whole mount in situ hybridization: WMISH) at P7 for quantification purposes, which has the advantage that quantifications can be

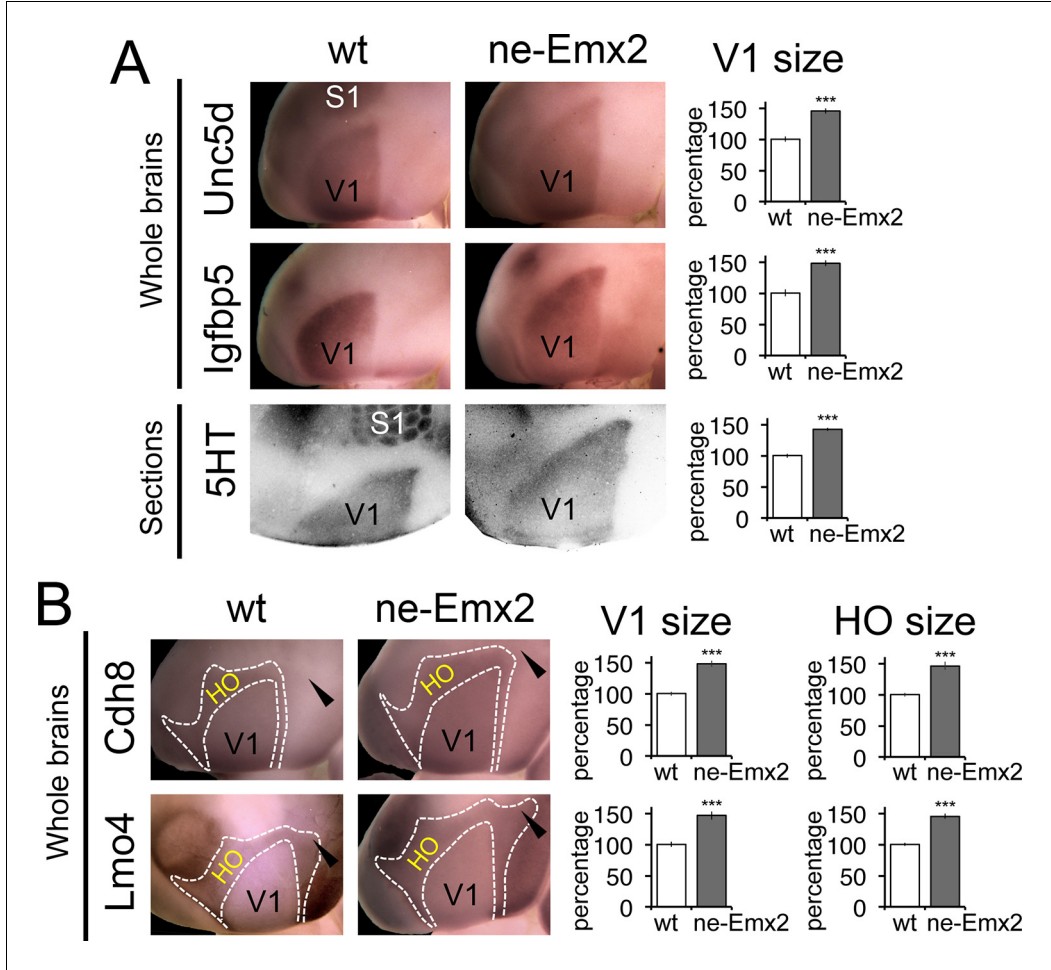

**Figure 3.** Proportionally increased V1 and HO sizes in *ne-Emx2* cortices. (**A**) Schematic shows sensory area outlines in the caudal neocortex (12). WMISH with the molecular V1 marker genes *Unc5d* (wt: n = 5, *ne-Emx2*: n = 6) and *Igfbp5* (wt: n = 11, *ne-Emx2*: n = 6) at P7 highlights increased V1 size in *ne-Emx2* brains using whole un-sectioned brains. Quantification of V1 size using 5HT-stained P7 flattened cortical sections similarly reveals larger V1 sizes in *ne-Emx2* brains (n = 11), compared to wt brains (n = 13). (**B**) WMISH for molecular markers that label both V1 and HO (dotted outlines: *Cdh8, Lmo4*: high expression in HO, lower in V1; for each probe and genotype n = 6) reveal that V1 as well as HO sizes in *ne-Emx2* are larger compared with wt brains. *Cdh8* is not expressed around the anteromedial edge of V1 (arrowheads). Quantifications in *Figures 3* and *4* show mean values as percent of wt, error bars indicate standard error of the mean; asterisks highlight statistical significance according to unpaired to t-test. 5HT, serotonin; S1, primary somatosensory cortex; WMISH, whole mount in situ hybridization; V1, primary visual cortex; wt, wildtype.

The following figure supplements are available for figure 3:

**Figure supplement 1.** *Lmo4* expression delineates primary and higher order cortical area boundaries in whole mount brains.

**Figure supplement 2.** *Lmo4* and *Cdh8* expression marks primary and caudal extrastriate cortical regions.

made using single images without sectioning and artifacts that may arise from such tissue processing. We first used a set of two marker genes, *Unc5d* and *Igfbp5*, whose expression delineates V1 at P7 (*Chou et al., 2013*). The gene expression domains on WMISH-stained brains were outlined and their sizes quantified as a measure of V1 area size. The mean value of wt brains was defined as 100% and the area size percentages of *ne-Emx2* brains displayed accordingly as 'percent of wt' (*Figure 3A*). V1 gene expression domains of both markers were larger in *ne-Emx2* brains (*Unc5d*-V1: 148% ± 6.1%, p = 0.0003; *Igfbp5*-V1: 148 ± 4.5%, p < 0.0001). The magnitude of the increased in V1 size labeled genetically in *ne-Emx2* brains is comparable to V1 area measurements derived from 5HT-stained P7 flattened cortical sections (*Figure 3A*: 5HT-V1: 142.1% ± 3.1%, p < 0.0001), indicating that molecular markers on whole brains reliably delineate V1 and can therefore be used to quantify and compare area sizes between samples and mouse lines.

We next have used additional markers to quantify higher order area sizes (*Figure 3B*). Previous studies have parsed higher order visual areas using neuroanatomical tracers (*Wang and Burkhalter, 2007*) (see also schematic in *Figure 3*) and have revealed genes that are expressed at different levels in V1 and higher order visual areas (*Chou et al., 2013*). For example, *Cdh8* and *Lmo4* expression is higher in the area surrounding V1, where higher order visual areas are located (*Chou et al., 2013*; *Marshel et al., 2011*; *Wang and Burkhalter, 2007*). The domains of high *Cdh8* and *Lmo4* expression appear to label higher order visual areas uniformly (*Chou et al., 2013*), without revealing subdivisions between them (compare schematic in *Figure 3* showing approximate location and outline of higher order visual areas as identified by *Wang and Burkhalter, 2007* to *Cdh8* and *Lmo4* gene expression domains around V1). On P7 WMISH-stained wt brains, we quantified the V1 and HO sizes in the medial cortex on the basis of low and high gene expression domains (see dotted lines in *Figure 3*, *Figure 3—figure supplement 2*). Anatomically, these gene expression domains surrounding V1, which show much stronger staining compared with V1, extend anteriorly up to the S1 border, laterally to the border of the auditory areas and the ECT and medially up to the border to the RSC, respectively (*Figure 3—figure supplements 1,2*). Hence, compared with the above-mentioned HO complex that was identified using 5HT staining (*Figure 1*), the higher order area complex labeled by *Cdh8* and *Lmo4* relates to a smaller cortical region that more closely relates to higher order visual areas, but excludes the RSC. The size (*Cdh8*-V1: 145.3 ± 7.4%, p < 0.0001; *Lmo4*-V1: 146.6% ± 6.7%, p = 0.0005) and shape of the gene expression domains in V1 were similar in *Cdh8*- and *Lmo4*-stained brains. Similarly, the gene expression domains nested around V1 largely overlapped between the two probes. The only apparent difference between them is around the anteromedial edge of the higher order visual areas (*Wang and Burkhalter, 2007*), where *Cdh8* is expressed at much lower levels compared to more lateral regions around V1 across genotypes (arrowheads in *Figure 3B*). The wt values of the measurements were again defined as 100%. The overall shapes of the two HO marker gene domains were similar and the sizes larger in *ne-Emx2* brains compared with wt brains (*Cdh8*- HO: 145.7 ± 6.4%, p = 0.0015; *Lmo4*- HO: 144.9 ± 3.8%, p = 0.00157). The analysis of different area-specific sets of marker genes, either showing unique expression in V1, or discernable expression levels between visual areas, revealed an increase in visually-related HO in *ne-Emx2* brains that was proportionate to the V1 size increase. The extrastriate areas that we have identified on the basis of 5HT staining (*Figure 1*) included the RSC, which is not a higher order visual area (*Garrett et al., 2014*; *Marshel et al., 2011*; *Vann et al., 2009*; *Wang and Burkhalter, 2007*) raising the possibility that only related primary and higher order areas (e.g. vision) could scale proportionately. To test this possibility, we have used WMISH of *Lypd1* on P7 wt and *ne-Emx2* brains as a specific marker labeling the caudomedial cortex, where the RSC is located (*Figure 3—figure supplement 1*). We found that the specific *Lypd1* gene expression domain in the caudomedial cortex is significantly enlarged in *ne-Emx2* brains (114.3 ± 5.2%, p = 0.0225, n = 4), compared with wt brains. This size increase is not proportionate to the size increases of V1 and the higher order visual area complex labeled by *Cdh8* and *Lmo4* in *ne-Emx2* brains (*Figure 3*) suggesting that increased V1 size is specifically accompanied by a proportionate size increase of related higher order visual areas.

To test if related HO size matches V1 size only when it is larger than normal, or if primary area size bi-directionally is accompanied by according scaling of related higher order areas, we next analyzed HO sizes excluding the RSC in brains with a smaller than normal V1 (*Figure 4*). Constitutive *Emx2* mutant mice have an overall smaller brain and visual cortex, but homozygous mutants die perinatally (*Bishop et al., 2000*), preventing the analysis of cortical areas, which arise at later stages. To overcome this limitation, we generated a novel mouse line with floxed *Emx2* alleles (*Figure 4—*

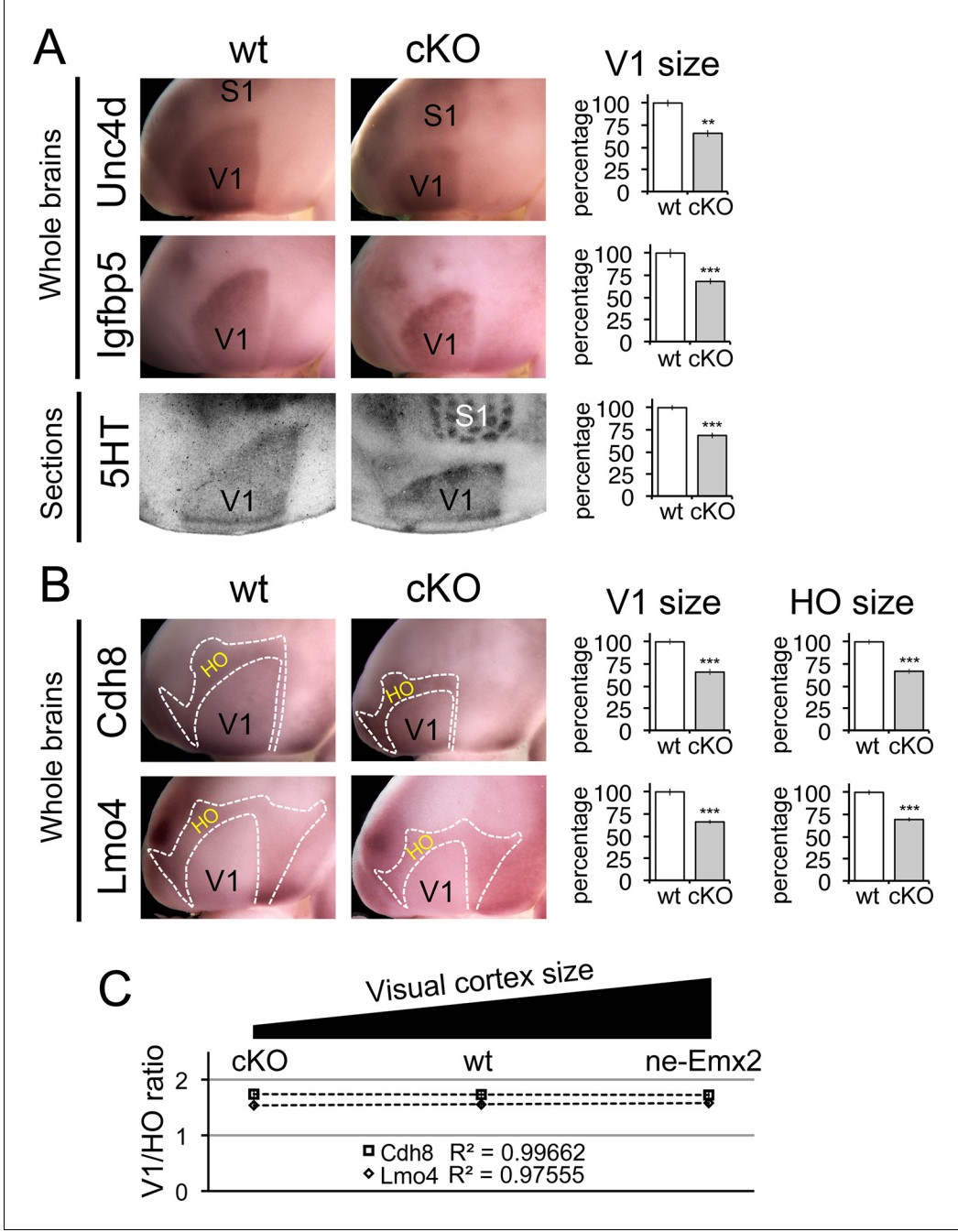

**Figure 4.** Proportionally decreased V1 and HO sizes in *cKO* cortices. WMISH for V1 (**A**: *Unc5d;* wt: n = 5, *ne-Emx2*: n = 6, *Igfbp5;* wt: n = 11, *ne-Emx2*: n = 8) or V1 and HO marker genes (**B**: *Cdh8;* wt: n = 6, *ne-Emx2*: n = 6; *Lmo4;* wt: n = 5, *ne-Emx2*: n = 5) at P7 conversely reveals decreased sizes (~70% of wt size) of V1 and HO in brains that were derived from *Emx1-IRES-Cre*-mediated cortex-specific conditional *Emx2* mutant brains (*cKO*), compared with wt brains. Quantification of V1 size using 5HT staining on P7 flattened cortical sections (A: wt: n = 15, *cKO*: n = 10) reveals similar reductions of V1 size in *cKO* brains. (**C**) The ratio between quantified V1 and HO sizes derived from WMISH-stained brains with decreased (*cKO*), normal (wt), and increased (*ne-Emx2*) V1 sizes demonstrates linear scaling of HO size in response to bi-directional changes of V1 size. S1, primary somatosensory cortex; V1, primary visual cortex; WMISH, whole mount in situ hybridization; wt, wildtype.

The following figure supplements are available for figure 4:

**Figure supplement 1.** Generation of *Emx2*-floxed mice and confirmation of cortex specific *Emx2* deletion.

**Figure supplement 2.** Normal cortical neuroanatomy in *cKO* brains.

*Figure 4 continued on next page*

*Figure 4 continued*

**Figure supplement 3.** Area patterning changes following cortex-specific deletion of *Emx2*.

**Figure supplement 4.** Posteriorly shifted V1 border in *cKO* cortices.

*figure supplement 1*), allowing conditional inactivation of *Emx2*. We crossed *Emx2 floxed* mice with *Emx1-IRES-Cre* expressing mice (*Gorski et al., 2002*) to generate conditional, cortex-specific *Emx2* mutant mice. These *cKO* mice are viable, fertile and have an anatomically normal neocortex (*Figure 4—figure supplement 2*). Confirming the prediction that reduced *Emx2* expression levels in cortical progenitors would lead to smaller visual areas, *cKO* brains show areal patterning changes (*Figure 4—figure supplement 3*) that are similar to those previously reported in heterozygous *Emx2* mutant brains (e.g. larger frontal cortex) (*Hamasaki et al., 2004*), but are opposite to those apparent in *ne-Emx2* brains (e.g. smaller frontal cortex) (*Hamasaki et al., 2004*; *Leingärtner et al., 2007*). As in *ne-Emx2* brains, V1 in *cKO* brains was characterized using 5HT staining and DiI injections into the dLG (*Figure 4—figure supplement 4*), revealing that V1 in *cKO* is greatly reduced in size.

To complement the quantification of V1 and its related HO sizes in *ne-Emx2* brains, we next have used *cKO* brains to perform WMISH with both sets of marker genes noted above (*Figure 3*) and measured their sizes (*Figure 4*). Measurements of the molecular V1 marker domains (*Unc5d*-V1: 67.1 ± 2.6%, p = 0.00029; *Igfbp5*-V1: 68 ± 3.3%, p = 0.00017), as well as the V1 expression domains of *Cdh8* (67.7 ± 4.5%, p < 0.0001) and *Lmo4* (68.6 ± 0.8%, p < 0.0001) revealed that the molecularly defined V1 in *cKO* was smaller than in wt brains. These reductions matched the reduced V1 in 5HT-stained flattened cortex sections (*Figure 4A*: 5HT-V1: 68.5 ± 3%, p < 0.0001). Subsequent quantification of the gene expression domains of *Cdh8* and *Lmo4* surrounding V1 (*Figure 4B*: *Cdh8*- HO: 66.8 ± 2.9%, p < 0.0001; *Lmo4*- HO: 69.3 ± 2.4%, p < 0.0001) revealed that the cortical region that contains visually-related HO was also reduced in *cKO* brains to a degree proportional to the reduction in V1 size. These data demonstrate that when V1 size is reduced, related HO size is reduced to a similar extent.

In order to reveal a correlation between primary and related higher order area size between brains with larger and smaller than normal visual areas, we calculated the ratios between the genetically defined V1 and related HO sizes (*Figure 4C*). Ranging over an ~80% variation of the normal V1 size, *Figure 4C* reveals that related HO size is bi-directionally altered in a linear fashion (*Cdh8* regression: y = −0.0071x + 1.7463; *Lmo4* regression: y = 0.0208 + 1.7463). Taken together, our results are consistent with a proportional scaling relationship between the size of primary and related higher order visual areas: The size of V1 is determined by the activity of transcription factors including *Emx2* during development, and this mechanism likewise controls the linear matching of the proportions of higher order visual areas in the mouse neocortex.

## Discussion

The present findings address the mechanisms that specify and regulate the size of higher order sensory areas, an issue that has been largely neglected. They reveal a novel, prominent role for intrinsic genetic information in this process. Genetically altering the size of V1 over a range of ~80% of its normal size using a *Emx2* gain-of function mouse line and a novel conditional *Emx2* loss-of function mouse line showed that the specification of both primary and related higher order cortical areas during development was linearly scaled by driving the unique properties that characterize both, V1 and higher order visual areas.

Regardless of whether V1 was larger or smaller than in wt mice, related HO exhibited normal cytoarchitecture, genetic profiles, functional properties, and characteristic patterns of connectivity that resulted in an overall uniformly altered 'visual cortical field' in the occipital cortex that remained accurately and proportionally subdivided into V1 and higher order visual areas. This demonstrates that *Emx2* (and perhaps additional intrinsic area patterning regulators) specify a 'sensory cortical field' that includes primary and higher order areas and a defined border between them. This model

of cortical area patterning is not consistent with the possibility that the core properties of primary and higher order areas are specified sequentially or through parallel genetic mechanisms.

Our results further reveal that higher order areas do not have a fixed size. Rather their relative size is flexible. By using mouse models with bi-directional changes of V1 size as a model, our study revealed that higher order areas scale linearly together with their related primary sensory areas, This observation is important for at least two reasons: (i) it re-emphasizes a sequential model of primary sensory area formation that likewise influences the properties of related higher order areas. In this model, cortical intrinsic mechanisms specify all generic primary and higher order visual cortex properties during early development. Much later during postnatal development, geniculocortical input is needed to terminally differentiate the genetic distinctions between V1 and HO (*Chou et al., 2013*; *Vue et al., 2013*). (ii) It contradicts the hypothesis that cortical structure/function evolution mainly is driven by a disproportionate increase in the size of related higher order areas relative to primary areas. To the contrary, our results show that primary and related higher order areas remain proportionate when primary area size is altered through genetic mechanisms, suggesting that an increase in the complexity of connections and micro-circuits among higher order cortical processing centers likely accounts for gains in cortical functions that are characteristic for gyrencephalic mammals with larger cortical surface areas, compared to simpler lissencephalic mammals. In summary, the newly discovered linear scaling relationship between primary and related higher order areas has major implications for the basic understanding of the development and organization of the neocortical bauplan and its evolution and variability in normal and affected conditions.

## Materials and methods

### Mouse lines and conditional *Emx2* gene targeting

All experiments were approved and conducted following the guidelines of the Institutional Animal Care and Use Committee at the Salk Institute and were in full compliance with the guidelines of the National Institutes of Health for the care and use of laboratory animals. When mice were mated, the morning of the identified vaginal plug was designated as E0.5. The morning on which pups were born was designated P 0.5. Transgenic mice overexpressing *Emx2* under the *Nestin* promoter (*ne-Emx2*) were previously described (*Hamasaki et al., 2004*). For generating *Emx2 floxed* mice (*Emx2fl/fl*), gene targeting was carried out using homologous recombination in embryonic stem cells. A targeting construct was designed in which the 5' *loxP* site was upstream of the *Emx2* transcriptional start site and the 3' *loxP* site downstream of Exon 2, followed by a *FRT*-site-flanked *PGK-Neo* cassette, *Figure 4—figure supplement 1*). Targeted embryonic stem cell clones were screened by Southern blot with probes A, B, and C and by PCR to identify *Emx2floxed-neo/+* clones (*Figure 4—figure supplement 1*). Positive clones were injected into C57BL/6J blastocysts at the Salk Transgenic Core Facility and chimeras were mated to C57BL/6J females to obtain germline transmission. Heterozygous mice were mated with mice expressing *FLPe* (*Rodríguez et al., 2000*) to remove the neo cassette and then mated to obtain homozygous *Emx2fl/fl mice*. Cortex specific deletion of *Emx2* (*cKO*) was obtained by crossing *Emx2fl/fl* mice with *Emx1-IRES-Cre* mice (*Gorski et al., 2002*). Specificity of *Emx1-IRES-Cre*-mediated deletion of *Emx2* floxed alleles was analyzed by WMISH (described below) staining using a full-length *Emx2* antisense RNA probe on E11 embryos. Genotyping was performed using primers for *Emx2* floxed alleles (*Emx2* forward: GAC-TCC-TTT-CCC-AAA-TAA-CCC-C, *Emx2* reverse: GTA-AGC-GGG-CGG-GGA-CTG-GTT-C) and for the *Cre* recombinase (cre forward: GCT-AAA-CAT-GCT-TCA-TCG-TCG-G, cre reverse: GAT-CTC-CGG-TAT-TGA-AAC-TCC-AGC), and the *ne-Emx2* transgene (nestin forward: TCA-ACC-CCT-AAA-AGC-TCC, Emx2 reverse: GGA-CGG-AGA-GAA-GGC-GGT).

### In situ hybridization, immunostainings, and tangential cortical sections

Tissues were dissected, washed in phosphate-buffered saline (PBS), fixed overnight in 4% phosphate-buffered paraformaldehyde (PFA), washed in PBS, and cryopreserved in 30% sucrose in PBS. Postnatal brains were perfused with PFA, postfixed overnight in PFA, washed with PBS, and cryopreserved in 30% sucrose in PBS. Tissues were embedded in Tissue-Tek OCT (Sakura Finetek , Japan) and sectioned on a cryostat (Leica, Germany). Antisense RNA probes were labeled using a DIG-RNA labeling kit (Roche, Switzerland). ISH on 18-μm cryostat sections and WMISH using P7 brains were

carried out as previously described (*Chou et al., 2013*; *Hamasaki et al., 2004*; *Zembrzycki et al., 2007*; *Zembrzycki et al., 2015*). For tangential cortical sections, cortical hemispheres were dissected, flattened, postfixed between slide glasses, and then cryoprotected. Tangential sections were cut into 40-μm slices from flattened cortical hemispheres with a sliding microtome and then they were immunostained for Serotonin (5HT, ImmunoStar, Hudson, WI). Immunostaining was developed using the diaminobenzidine colorimetric reaction and the Vectastain kit (Vector, Burlingame, CA). For Nissl staining, sections were stained with 0.5% cresyl violet and then dehydrated with graded alcohols.

## Axon tracings

Lipophilic tracers DiI and DiD (all from Molecular Probes, Eugene, OR) were used to label corticothalamic-, thalamocortical-, corticotectal-, and corticospinal projections. For each experiment 4–6 brains with comparable tracer injection sites were cut and used for further data analysis, representative example images are shown in the figures. Analysis of thalamocortical axons by thalamic DiI injections (*Figure 1B*): P7 brains were fixed in 4% PFA, hemisected, and a coronal cut between the diencephalon and mesencephalon was made in order to expose thalamic nuclei at the section surface and DiI crystals were implanted to cover the dorsal lateral geniculate nucleus (dLG). After incubation for 1 to 2 months at 30°C to 60°C, preparations were sectioned sagittally on a vibratome (Leica). Sections were counterstained with DAPI (Vector) and analyzed under a fluorescence microscope to determine the tangential distribution of labeled thalamocortical axons in the neocortex.

Analysis of layer 5 subcerebral projection neurons (*Figure 1C*): Corticospinal neurons in cortical layer 5 were retrogradely labeled by inserting DiI crystals into the pyramidal decussation in 4% PFA fixed brains. Layer 5 corticotectal neurons were labeled in 4% PFA fixed brains by implantation of small DiI crystals into the upper layers of superior colliculus. Brains were incubated at 37°C for 2–3 months before 100 μm sagittal vibratome sections were cut and analyzed under fluorescent light. Analysis of area-specific thalamocortical and corticothalamic connectivity of caudal cortex (*Figure 2*): P7 pups were anesthetized by hypothermia and a small area of skull was removed to expose the cortical surface. DiI crystals and a small piece of DiD were implanted into cortical locations around V1 and the V1/HO border. After 1 day of survival, brains were removed after 4% PFA perfusion and their cortices and thalami dissected. Cortices were then flattened and stained for 5-HT to reveal primary sensory areas relative to the dye injection sites. The thalami were preserved sectioned coronally, stained with DAPI and DiI/DiD labeled cells analyzed under a fluorescence microscope.

## Statistical analysis, area measurements, and intrinsic signal optical imaging

Data collection and analyses were performed blind to genotype and the conditions of the experiments, data were collected and processed randomly, and no data points were excluded. No statistical methods were used to predetermine sample sizes, but our sample sizes were similar to those reported in previous publications (for example, (*Chou et al., 2013*; *Zembrzycki et al., 2013*). Data met the assumptions of the statistical tests used, and the data distribution was assumed to be normal but was not formally tested. Statistics were calculated with Microsoft Excel. Quantifications show mean values of the tested groups and are displayed as a percentage of the wt group. Quantified sample sizes (number of brains: n) are indicated in the figure legends. The examples shown in each figure are representative and were reproducible for each set of experiments. Individual experiments were successfully repeated at least three times using different litters.

Area size measurements on 5HT stained sections and statistical analysis was performed as previously described (*Leingärtner et al., 2007*; *Zembrzycki et al., 2015*; *2013*). To quantify molecular V1 and HO sizes, gene expression domains were quantified on single images of WMISH-stained brains (examples of measured area outlines are shown as dashed lines in *Figures 3,4*) using ImageJ (Rasband 1997−2013). Derived wt mean values were defined as 100% and values of the other mouse lines calculated accordingly. Statistical significance was determined using unpaired two-tailed t test, p values < 0.05 (indicated as *) were considered as statistically significant. Variance is indicated in the main text sections reflecting standard error of the mean. Intrinsic signal optical imaging was performed as previously described (*Kalatsky and Stryker, 2003*). To determine the spatial relationship between V1 functional maps and V1, defined histochemically by 5-HT staining, animals were imaged

to determine the V1 map, and after completion, small DiI injections were made outside of the functional map perimeters. Animals were then perfused with 4% PFA, cortices dissected, flattened, sectioned tangentially, and stained for 5-HT.

## Acknowledgements

We thank Kevin Jones for providing Emx1-IRES-CRE mice. This work was supported by the Vincent J Coates Chair of Molecular Neurobiology at the Salk Institute for Biological Sciences (DDMO).

## Additional information

### Funding

| Funder | Grant reference number | Author |
| --- | --- | --- |
| National Institutes of Health | NS31558 | Dennis DM O'Leary |
| National Institutes of Health | MH086147 | Dennis DM O'Leary |
| National Institutes of Health | EY02874 | Michael P Stryker |

The funders had no role in study design, data collection and interpretation, or the decision to submit the work for publication.

### Author contributions

AZ, Conception and design, Acquisition of data, Analysis and interpretation of data, Drafting or revising the article; AMS, Acquisition of data, Analysis and interpretation of data, Drafting or revising the article; AL, SS, SJC, VK, SRM, Acquisition of data, Analysis and interpretation of data; MPS, Analysis and interpretation of data, Drafting or revising the article; DDMO'L, Conception and design, Analysis and interpretation of data, Drafting or revising the article

### Author ORCIDs

Andreas Zembrzycki, http://orcid.org/0000-0001-8468-4195

### Ethics

Animal experimentation: All experiments were approved under Protocol #09-012 and conducted following the guidelines of the Institutional Animal Care and Use Committee at the Salk Institute and were in full compliance with the guidelines of the National Institutes of Health for the care and use of laboratory animals.

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
