## [Decision Letter]

Thank you for submitting your work entitled "Genetic mechanisms control the linear scaling between cortical primary and higher order sensory areas" for consideration by *eLife*. Your article has been reviewed by three peer reviewers, and the evaluation has been overseen by a Reviewing Editor and David Van Essen as the Senior Editor.

The reviewers have discussed the reviews with one another and the Reviewing editor has drafted this decision to help you prepare a revised submission.

The three reviewers found this study of visual cortical size regulation to be cleverly designed and of interest. The manuscript revisits earlier studies on the regulation of higher visual cortex specification and provides new insights into the genetic basis of cortical development.

Several major issues were brought up by the reviewers. One concern is the way the topographic locations of VI and how the parcellations of VHO were defined. It was felt the effects were not conclusive due to potentially imperfect flat-mounting of cortex and inconsistent delineation of V1 and higher order cortex. Another related reservation is the lack of evidence for reliable areal parcellation. Although the experimental design of manipulating the size of V1 is clever and elegant, without reliable parcellation of the cortex, the results fall short of being convincing to demonstrate that the size of V1 is scaled and that there is a clear cause and effect between a change in the size of V1 and the size of neighboring higher order areas.

The referees raised several specific issues that were deemed essential. They requested that the revised manuscript demonstrate that: (1) the size of V1 in control and experimental brains was determined in identical fashion; (2) that the measurements of the borders of VHO are clearly outlined in control and experimental brains; (3) The borders of the extrastriate visual cortex need to be better defined. Does VHO exclude (or include) retrosplenial cortex; and (4) the lateral border of VHO relative to primary auditory cortex, ventral posterior temporal cortex and rhinal fissure needs to be clearly defined. More detailed comments directly related to these concerns are elaborated below with recommendations for improvement:

1) The VHO in the EMX2 case includes retrosplenial cortex and extends deeper into temporal cortex than in WT animals. This raises the questions how the borders of VHO were defined and what types of criteria for flattening the cortex were used to compare the sizes of V1 and VHO. There is strong evidence that VHO extends far beyond extrastriate visual cortex. Thus, one option is to drop the "V" and call the region "HO".

2) Figure 1. Labeling of the thalamocortical input is challenging. It is difficult to rule out that labeling patterns are produced by injections at particular topographic locations. The experiment should indicate the LGN injection site and the projections in flatmounted cortex, superimposed onto 5HT labeled V1 in the same mouse.

3) Figure 1. The analysis could be strengthened by 5HT staining of parallel sections showing S1 and V1.

4) Figure 2. A more compelling demonstration of the change in the size of V1 and VHO would be to mark reference points in the topographic map, stain the same flattened cortex for 5HT and superimpose the images. Further, it would strengthen the interpretation of the data if the maps included information about the topographic representations in areas of the lateral extrastriate cortex.

5) Figure 2. The cartoon of the mouse thalamus and the labels are inaccurate for the DiI and DiD marked projections. The DiI and DiD injections into WT and *ne-Emx2* mice are targeted to different topographic locations of V1 and areas within VHO. These discrepancies weaken the conclusion that the corticothalamic projections are comparable.

6) Figure 3. The borders of VHO in WT and *ne-Emx2* mice are different. Again VHO is ill-defined and its borders are not consistently outlined. For example, the *Cad8* (WT) stained region around the tip and continuing along the medial border of V1 is much larger than indicated by the stippled line. In addition, V1 in *ne-Emx2* mice has an unusual shape with an unexpected "tipped nose", indicating that the region enclosed by the stippled line may be larger than V1. The flattening technique can contribute to an overestimation of the size differences reported in the study.

7) Figure 4, similar to Figure 3. *Cad8* (WT) and *Lmo4* (WT) include weakly stained regions in medial extrastriate cortex, which are not included in *cKO* mice.

[Editors' note: further revisions were requested prior to acceptance, as described below.]

Thank you for resubmitting your work entitled "Genetic mechanisms control the linear scaling between cortical primary and higher order sensory areas" for further consideration at *eLife*. Your revised article has been evaluated by David Van Essen (Senior Editor), a Reviewing Editor, and two reviewers. The manuscript has been improved but there are some remaining issues that need to be addressed before acceptance, as outlined below:

The two prior reviewers indicated the paper has been improved by additional 5HT data to address the location of V1 with other cortical regions and with the additional analysis of the borders of adjacent areas by gene markers, *Lmo4* and *Cad8*. The modifications and explanations in the revision help to clarify the questions that were previously raised. The new images that are provided are impressive and strengthen the study.

However, there is still a reservation about how V1 and HO are identified and defined. Re-evaluation of the figures indicates that the boundaries of HO differ from one figure to another. Specifically, in Figure 1, the border of HO runs from the anterior retrosplenial cortex along the border of auditory cortex and ends near the rhinal fissure. But in Figure 3, HO does not extend to the border of auditory cortex or include the retrosplenial cortex. Hence, there are two different criteria used to define HO in the manuscript. Furthermore, *Cdh8* in WT and *ne-Emx2* extends more medially than indicated in Figure 3 and also in Figure 4, leading to a discrepancy in the analysis.

It is strongly recommended that a re-analysis of the data take place with a critical eye to establish a more consistent segmentation scheme for defining the detailed borders of HO. One suggestion is to measure HO as it is presented in Figure 1 and to exclude retrosplenial cortex. The resulting region would represent a large amount of visually-related territory.

The previous review indicated there was not enough evidence to conclude that V1 size controls the sizes of surrounding cortical areas. Therefore, the Abstract should be modified to emphasize that genetic factors intrinsic to the cortex proportionately dictate the size of primary visual cortex and surrounding higher order areas.

Reviewer #1:

The authors have very largely addressed the suggestions of the previous review, and their paper now seems ready for publication.

One remaining point: The Abstract needs to be slightly rewritten to bring it into line with the rest of the paper.

In the Abstract it is stated: "we have generated bi-directional changes in primary visual cortex size in vivo and have used it as a model to show a novel and prominent function for genetically determined primary visual area size, which also proportionally dictates the sizes of higher order visual areas." It seems (again) that the authors want to say that V1 size controls the size of surrounding higher order areas. Previous review suggested there was not enough evidence for such a conclusion, and pointed to an alternate interpretation: genetic factors intrinsic to cortex control the size of both V1 and HO areas in coordination. The cover letter for the revised MS and the new Discussion section appear to agree with the latter conclusion.

Reviewer #3:

This is a reply to comments about how primary visual cortex and the surrounding higher order cortex were delineated and the relative size of these cortical parcels were measured and compared. The detailed reply indicates that the authors clearly understood the critique and made a number of revisions in the figures and the text that significantly improved the manuscript. Although V1 and the organization of surrounding cortex are beautifully illustrated in several newly added images, there are remaining concerns that the revision failed to analyze the outstanding material in a consistent fashion. This impression is based on the inexplicable discrepancy of the definition of HO shown in Figure 1 and Figure 3, which indicates that the authors are unsure how to subdivide the real estate. In Figure 1 it appears that the border of HO runs from the anterior tip of retrosplenial cortex along the posterior border of S1, continues along the border of auditory cortex and ends by intersecting the rhinal fissure. This does not fit the scheme shown in Figure 3, in which HO does not extend to the border of auditory cortex and does not include retrosplenial cortex. If I understand this correctly, HO in Figure 3 is clearly smaller than in Figure 1 and does not include ventral posterior temporal cortex nor does it include retrosplenial cortex. In fact, HO Figure 3 is what was measured in Figure 3 and Figure 4. Maybe these discrepancies were simply an oversight which can be easily corrected. But the lapse heightened my concerns that the size of HO was inconsistently assessed (e.g. in Figure 3 the border of *Cdh8*-expression in wt is much more medial than indicated and differs from the border in *ne-Emx2*) and that the substance of the paper is less solid than suggested by the graphs shown in Figure 3 and Figure 4. I recommend re-analyzing the material with a more critical eye on the detailed borders of HO. I am sure that the authors are acutely aware that details matter a lot more in small than large brains.

[Editors' note: further revisions were requested prior to acceptance, as described below.]

Thank you for resubmitting your work entitled "Genetic mechanisms control the linear scaling between related cortical primary and higher order sensory areas" for further consideration at *eLife*. Your latest revision has been favorably evaluated by David Van Essen (Senior Editor), a Reviewing Editor, and the two prior reviewers. There are some remaining issues that need to be addressed before acceptance, as outlined below:

Reviewer #3:

I am encouraged by the revision in paragraph seven, Results, which now more clearly defines the borders of HO. Several recommendations are made to clarify the following points.

1) To resolve the conflict between the measured HO (Figure 3, Figure 4) and HO defined by 5HT staining (or the virtual absence of 5HT expression) in Figure 1 suggest renaming it HO-5HT to indicate the conditions under which the observation was made and unequivocally state that HO and HO-5HT are not the same.

2) HO includes at a minimum visually dominated as well as higher order auditory areas. I therefore recommend revising the sentence (Abstract) to: "…surrounding higher order sensory areas". I further recommend that similar revisions be made throughout the text to make it absolutely clear that the cortex measured here is more extensive than extrastriate visual cortex.

3) I recommend revising the border of HO (wt/*Cdh8*) in Figure 3. It is clearly evident in the image that the *Cdh8* expression includes a wedge medial to V1 which extends to the tip of the arrow head. A similar adjustment of the medial border is necessary for *ne-Emx2 Cdh8* (Figure 3). The same revisions are necessary in Figure 4 in which the medial borders of *Cdh8* expression in wt and cKO are more medial than indicated.

---

## [Author Response]

*The three reviewers found this study of visual cortical size regulation to be cleverly designed and of interest. The manuscript revisits earlier studies on the regulation of higher visual cortex specification and provides new insights into the genetic basis of cortical development.*

*Several major issues were brought up by the reviewers. One concern is the way the topographic locations of VI and how the parcellations of VHO were defined.*

To address the concern raised by the reviewers regarding how the topographic locations of V1 were defined, we have revised the manuscript to better explain our approach and results. In general in Figure 1 and Figure 2, we introduce V1 along with the other cortical areas on the basis of 5HT staining that labels thalamocortical input, with area-specific marker genes like *Rorb* and by using functional imaging. All of these approaches have been used in many studies from many different labs over many years with consistent results. We feel, for example, that there is no doubt that the triangular-shaped 5HT domain in the caudal cortex (Figure 1) is V1. Similarly, we do not hesitate to state that the functional imaging maps (Figure 2) reveal V1 and its topography. To address the reviewer’s concern, we have added 5HT images showing the entire cortical hemisphere into Figure 1 to better illustrate the location of V1 and all the other apparent cortical areas and regions on the entire cortical sheet. This now demonstrates much more clearly where, for example, the border between neocortex and entorhinal cortex is located and that 5HT staining is not apparent in the retrosplenial cortex. On the adjacent higher magnification images, we then demonstrate that extrastriate areas including retrosplenial cortex appear enlarged in *ne-Emx2* brains qualitatively. We have revised the text accordingly to implement this information and to better define HO (paragraph one, Results).

In regard to the parcellation of HO in Figure 1, we have revised the text accordingly. As it now is more clearly demonstrated on the low magnification images, HO areas are now defined as follows: “We have defined this caudal cortical territory, which lies outside of V1, S1, and Aud and shows no or weak 5HT staining as a joint higher order cortical area complex (HO). Based on the 5HT staining, this region contains the higher order visual areas surrounding V1 (Wang and Burkhalter 2007), the retrosplenial cortex medially, as well as the ventral posterior temporal cortex laterally.”

In this study, we specifically avoid making claims about parcellation within higher order visual areas, since to our knowledge no marker genes exist that would allow to conclusively distinguish the at least 7-9 (Marshel et al. 2011, Wang and Burkhalter 2007) different higher order visual areas from one another.

It was felt the effects were not conclusive due to potentially imperfect flat-mounting of cortex and inconsistent delineation of V1 and higher order cortex.

We agree with the reviewers’ comments that flat-mounting potentially can have an effect on the measurement or interpretation of the data. However, we do not think that such artifacts have a major influence on the conclusions of our study for several reasons. Except for Figure 1, we mostly use sagittal sections or un-sectioned whole-mount brains (Figure 3 and Figure 4). This is especially true for the quantifications of V1 and HO that were exclusively derived from images of unsectioned whole brains. In all other instances where we use flattened cortical sections, we only make qualitative claims about the relative area proportions (e.g. V1 is bigger or smaller). We emphasize that all specimens were processed identically which suggests that possible artifacts would affect control and experimental cases equally, still allowing comparisons between brains. For example, it is very common that the staining of the auditory areas using 5HT staining on flattened sections is a bit fuzzy, while area borders more medially in the cortex are almost always very sharp (e.g. barrels). As we all know the auditory area borders are in reality as sharp as the borders of the more medially located areas. This phenomenon is due to the curvature of the cortex that is highest laterally in the cortex, where auditory areas are located. Flattening artifacts are therefore strongest laterally, where the cortex is curved the most, which consistently results in somewhat fuzzy staining around the lateral most cortical edge. Fuzzy borders of the auditory areas are indeed apparent in flattened sections in Figure 1. Importantly the fuzziness of the staining still allows for reliable delineation of the auditory areas, and the staining is comparable between wt and *ne-Emx2* sections. Further, we compare the quantification of V1 gene markers using un-sectioned whole brains with quantification of V1 derived from 5HT-stained flattened tangential sections (paragraphs six and nine, Results). The quantified sizes of the 5HT-stained V1 were in all cases very similar to the quantifications using marker genes, suggesting that the flattening and staining procure used in this study delivered reproducible results and contained no major artifacts due to the flattening procedure. To emphasize this point more clearly, we have moved the images and graphs of the quantified 5HT-stained flattened sections that were originally figure supplements into Figure 3 and Figure 4, respectively. Overall we believe that the use of whole-mount and well as sectioned data excludes the possibility that flattening artifacts alter the interpretation of our findings and our conclusions.

We discuss the issue about possible inconsistent delineation of V1 and HO in the comment below.

*Another related reservation is the lack of evidence for reliable areal parcellation. Although the experimental design of manipulating the size of V1 is clever and elegant, without reliable parcellation of the cortex, the results fall short of being convincing to demonstrate that the size of V1 is scaled and that there is a clear cause and effect between a change in the size of V1 and the size of neighboring higher order areas.*

Apart from the qualitative assessments of V1 and HO in Figure 1–Figure 2, we also report a quantitative assessment of the sizes of V1 and HO in Figure 3 and Figure 4. For this purpose, we use a panel of marker genes that either shows expression domains in V1 alone (Unc5d, Igfbp5) or also in HO (*Lmo4, Cad8*). All of the reported marker genes have been used successfully in previous studies to assess the size and location of primary areas including V1 (Cholfin and Rubenstein 2007, Chou et al. 2013, Hamasaki et al. 2004, Leingartner et al. 2003, Leingartner et al. 2007, Zembrzycki et al. 2015). Our study is novel in introducing *Lmo4* and *Cad8* as markers that also can be used to delineate and to quantify HO. We acknowledge that the reviewer’s comment in this regard was appropriate for the original manuscript. We have therefore added a new figure supplement (Figure 3—figure supplement 2) to the manuscript that better illustrates and identifies area borders on the basis of *Lmo4* and *Cad8* expression in wt and *ne-Emx2* brains. In this new figure we show whole brains in dorsal view that were stained in situ for *Cad8* or *Lmo4*. We present two sets of images for wt and *ne-Emx2* brains: One showing the raw images and another set showing the in situ images with superimposed annotated area borders on the basis of the marker gene expression domains. We think that these images much better illustrate how area borders were identified and used in this study to quantify V1 and HO sizes in Figure 3 and Figure 4. In addition to the representative cases that are shown on higher magnification in Figure 3 and Figure 4, the new figure supplement provides the reader with an additional set of marker gene-stained whole brains that show comparable area outlines on the basis of marker gene expression domains. We think that incorporating these additional cases provides additional evidence on the reliability and reproducibility of this technique to quantify V1 and HO.

*The referees raised several specific issues that were deemed essential. They requested that the revised manuscript demonstrate that: (1) the size of V1 in control and experimental brains was determined in identical fashion;*

We emphasize that all specimens were processed identically and the sizes of V1 determined identically on the basis of 5HT staining patterns, area-specific marker gene expression domains, or intrinsic functional response properties, respectively. We again apologize for the inconsistencies in some of the originally presented areal delineations. As we explain in more detail in our response to related comments, we have revised the representative outlines where they were inaccurate.

*(2) That the measurements of the borders of VHO are clearly outlined in control and experimental brains;*

We apologize for the inconsistencies in the area outlines in the original version of the manuscript. As we have detailed in our response to the specific comments below, we have carefully revised the annotations and have also added a new data as Figure 3—figure supplement 2 that details the gene expression domains of *Cad8* and *Lmo4* on views of the entire brain and how they relate to area borders in the cortex that we have used for area quantification.

*(3) The borders of the extrastriate visual cortex need to be better defined. Does VHO exclude (or include) retrosplenial cortex;*

The reviewers’ concern results from the failure of the original manuscript to provide some necessary details on the definition of cortical areas. In Figure 1, we define primary sensory areas and major cortical subdivisions on the basis of 5HT staining on flattened sections of the cortical hemisphere. 5HT is strongly expressed in primary areas V1, S1, auditory areas. A clear change to much lower 5HT staining in the entorhinal cortex makes it possible to identify the border between neocortex and entorhinal cortex laterally. The caudomedial cortical pole is devoid of 5HT staining; hence no distinction between higher order visual areas around V1 and the retrosplenial cortex around the cortical midline can be made on the basis of 5HT staining. Therefore in the qualitative assessment (Figure 1) of the region that lies outside of the primary sensory areas, the retrosplenial cortex is included. To better illustrate cortical parcellation and how they are affected in *ne-Emx2* brains, we have added new images showing 5HT staining on the entire cortical hemisphere into Figure 1. The higher magnification set of images that focuses on extrastriate areas and how they appear to expand qualitatively in *ne-Emx2* brains follow after them. We have also revised the text accordingly to now more precisely define the different cortical regions and how we have defined their borders (paragraph one, Results).

In addition, in response to the reviewer’s comment on the borders of HO, and we now present images of the genetic markers *Lmo4* and *Cad8* and note that the quantitative assessment of V1 and HO sizes were made without including the retrosplenial cortex medially. We have added according statements into the text (paragraph seven, Results) to describe these details more comprehensively.

*(4) The lateral border of VHO relative to primary auditory cortex, ventral posterior temporal cortex and rhinal fissure needs to be clearly defined.*

We agree that a comprehensive demonstration of how 5HT and marker gene staining relates to these boundaries was missing. To better define these borders using 5HT staining on flattened cortical sections, we have introduced new 5HT images showing the entire cortical hemisphere (Figure 1). We have also modified the text accordingly to include these definitions (paragraph one, Results).

In addition, to define these borders more clearly in un-sectioned whole-mount brains stained in situ, we have introduced a new figure supplement (Figure 3—figure supplement 1). This figure shows how borders to the auditory cortex and entorhinal cortex were defined through the use of *Lmo4* as a marker gene. Due to the curvature of cortex, not all cortical subdivisions are visible from every angle. This is especially true for the auditory cortex and entorhinal cortex in dorsal view. By using three different angles (lateral view, dorsolateral view, dorsal view) of the same *Lmo4*-stained brain, the new figure supplement now illustrates in raw images and annotated images how sharp gene expression borders were used to define subdivisions of the cortex. For the quantification of HO using marker genes shown in Figure 3 and Figure 4, we have exclusively chosen to use images of the dorsal view, since it is the best suited angle that gives the most-complete view of HO, while still allowing to define the borders to the auditory areas, the entorhinal cortex and the retrosplenial cortex in a single image. We accordingly now refer to this new data in the text (paragraph seven, Results).

This new figure supplement also adds a demonstration of the expression of the gene *Lypd1* to the manuscript. We have found that, *Lypd1* is not significantly expressed in V1 and HO, but strongly labels the retrosplenial cortex on whole mount images in the dorsal view (Figure 3—figure supplement 1). The gene expression differences of *Lmo4* between HO and retrosplenial cortex are more subtle, compared to *Lypd1* expression, but detectable (Figure 3—figure supplement 1). By comparing the expression of *Lmo4* and *Lypd1* in the medial cortex, this figure supplement demonstrates that the subtle *Lmo4* gene expression differences in this region reliably reveal the subdivision between HO and the retrosplenial cortex.

*More detailed comments directly related to these concerns are elaborated below with recommendations for improvement:*

*1) The VHO in the EMX2 case includes retrosplenial cortex and extends deeper into temporal cortex than in WT animals. This raises the questions how the borders of VHO were defined and what types of criteria for flattening the cortex were used to compare the sizes of V1 and VHO. There is strong evidence that VHO extends far beyond extrastriate visual cortex. Thus, one option is to drop the "V" and call the region "HO".*

We agree with the reviewers’ comment to more conclusively define the borders of V1 and VHO based on 5HT staining in Figure 1. Therefore, we have modified Figure 1 accordingly. We now show 5HT staining also on the entire cortical hemisphere and describe how the borders of 5HT staining in layer 4 reveals major anatomical subdivisions of the cortex including primary sensory area borders and laterally the border from the neocortex to the entorhinal cortex. We have also revised the text accordingly (paragraph one, Results). Further we have followed the reviewers’ suggestion and have dropped the term VHO and instead refer more broadly to higher order areas (HO) throughout the text and figures and now state that the delineation of ‘HO’ in Figure 1 contains higher order visual areas and the retrosplenial cortex, since 5HT staining does not reveal the subdivision between them.

*2) Figure 1. Labeling of the thalamocortical input is challenging. It is difficult to rule out that labeling patterns are produced by injections at particular topographic locations. The experiment should indicate the LGN injection site and the projections in flatmounted cortex, superimposed onto 5HT labeled V1 in the same mouse.*

We agree that labeling of TCAs can be challenging, but similar experiments are very common in the field and have been utilized to provide reproducible results. Although it is difficult to rule out entirely that differences in staining patterns in the cortex are influenced by tracer injections into different topographic locations in the thalamus, we believe that the presented DiI stainings provide very solid evidence for the claims made (Figure 1—figure supplement 1): (i) TCAs from the dLG extend more anteriorly in *ne-Emx2* brains. (ii) The border between DiI-stained and un-stained cortical tissues remains sharp in *ne-Emx2* brains. We believe that these claims are well supported by the presented data, because we observe these staining patterns consistently as demonstrated by examples on five different levels that cover the cortex from medial to lateral. This indicates that the filling of the dLG with dye was broad, robust, and comparable across genotypes. Further, the results of the DiI labeling from the dLG are consistent with an enlarged 5HT-stained V1. We remain confident that expansion of the DiI-stained domains at all tested medial to lateral levels in *ne-Emx2* brains are reflecting an enlargement of V1 that is evident using 5HT staining by using an alternative approach.

However, we have made some adjustments to the presentation of this data in the manuscript: We have removed the exemplary single DiI image in Figure 1 and instead in the main text now exclusively refer to the entire set of five different levels that are shown in Figure 1—figure supplement 1 (paragraph one, Results). As suggested by the reviewer, we have also added representative images that show strong and complete filling of the dLG with DiI in the sagittal plane (Figure 1—figure supplement 1, panel B) demonstrating that the filling of the dLG with dye in wt and *ne-Emx2* cases is comparable and robust.

Further, we feel that suggested images of DiI-stained, flat-mounted cortical sections with or without 5HT staining would not provide more directly relevant information to support the claims made. After all, we have decided not to introduce such images mainly because the laminar distribution and borders between stained and unstained cortical regions could potentially be distorted due to the flattening procedure. This issue can be circumvented and the laminar DiI distribution clearly demonstrated using sagittal sections as shown in Figure 1—figure supplement 1.

*3) Figure 1. The analysis could be strengthened by 5HT staining of parallel sections showing S1 and V1.*

We agree that 5HT staining complementing the apparent shift in distribution of layer 5 output projection neurons in *ne-Emx2* sections would be ideal. We would like to emphasize that this experiment requires an extremely long incubation time (4-6-month incubation in PFA at 37°C) in order to achieve sufficient in-vitro retrograde dye-labeling in the cortex from the far out injections sites in the pyramidal decussation and superior colliculus, respectively. In some pilot experiments we indeed generated such dual DiI labelings and performed 5HT stainings after the necessary diffusion time of the tracers using similarly processed specimens. Contrary to clear DiI-staining distributions in the cortex, we never obtained reliable 5HT staining in these brains. Proper immunostainings in these specimens are likely hampered by a combination of many month-long, extreme PFA over-fixation and prolonged elevated storage temperature over a substantial period of time.

Because of the low projected success rate and the fact that the anterior enlargement of 5HT staining in V1 layer 4 in *ne-Emx2* brains is very well documented (Hamasaki et al. 2004, Leingartner et al. 2007), we did not intend to perform 5HT staining on consecutive sections for this study. Overall we think that the presented DiI data clearly reveals that in layer 5 in *ne-Emx2* brains the balance of corticotectal versus corticospinal projections is altered in a way that is consistent with an anterior expansion of V1 and higher order areas: Qualitatively, *ne-Emx2* brains have an enlarged region sending out corticotectal projections, to the expense of a smaller region that send out corticospinal projections.

*4) Figure 2. A more compelling demonstration of the change in the size of V1 and VHO would be to mark reference points in the topographic map, stain the same flattened cortex for 5HT and superimpose the images.*

The present Figure 2 is exactly the experiment suggested, in which we marked the borders of V1 in vivo during the creation of the functional maps and then located the marks post mortem on sections stained for 5HT. Fourier imaging reveals the intrinsic functional neuronal properties of neurons in response to a visual stimulus that was moved across the visual field, from left to right and top to bottom and has been shown to reveal the borders of the primary visual area in the mouse (Kalatsky and Stryker 2003).

We should also note that each of the color-coded regions in the response maps represent visual responses to stimulation around a certain region of the visual field. Regions of similar color in different maps can serve as a reference point or reference region for areas responsive to the same portion of the visual field. In the text we now note that the green region representing ~20 to ~30 degrees of elevation/azimuth is clearly enlarged in *ne-Emx2* brains, compared to wt brains (paragraph four, Results).

*Further, it would strengthen the interpretation of the data if the maps included information about the topographic representations in areas of the lateral extrastriate cortex.*

While it might be interesting to demonstrate the complete parcellation of HO into many separate functional areas, doing so would be far beyond the scope of the present manuscript. We chose not to attempt this with the functional maps because we had, and still have, no corresponding genetic parcellation. In addition, the stimuli used for our functional maps like those in Figure 2 were not ideal for demonstrating the organization of all of the extrastriate cortical areas. Hints of them can be seen in at the lower left in the two leftmost panels of Figure 2, but no attempt was made to delineate them all, as has now been done for many of them in several laboratories (e.g.: Andermann et al. 2011, Garrett et al. 2014).

*5) Figure 2. The cartoon of the mouse thalamus and the labels are inaccurate for the DiI and DiD marked projections. The DiI and DiD injections into WT and ne-Emx2 mice are targeted to different topographic locations of V1 and areas within VHO. These discrepancies weaken the conclusion that the corticothalamic projections are comparable.*

We should note that the color-coding of the schematics does not refer to the dye injection sites shown in the original Figure 2 (now Figure 2). The colors were intended to provide readers that are not familiar with the stereotypic connections between thalamus and cortex with a visual aide using matching colors of the cortical areas and their corresponding thalamic regions to illustrate that S1 is interconnected with the VP, V1 with the dLG, and HO with the PO, respectively. The fact that the reviewers found this color-coding to be confusing led us to remove the color-coding from the schematics in Figure 2. They now are in black and white.

Further, we would like to clarify that it was by design that the injections were made at different topographic locations. By doing so, we aimed to investigate whether the actual connectivity patterns were consistent with the predicted patterns of connectivity based on the 5HT staining around the injection sites. The results indicate just that: Following overall changes in V1 size in *ne-Emx2* brains, the area borders of V1 and HO shift anteriorly. This demonstrates that although parts of V1 and HO in *ne-Emx2* brains are located at different topographic locations on the cortical sheet compared to wt, the neurons show connectivity patterns that are consistent with their altered intrinsic areal identity, not their topographic location on the cortical sheet. We have revised the text section to emphasize this issue more clearly (paragraph five, Results).

*6) Figure 3. The borders of VHO in WT and ne-Emx2 mice are different. Again VHO is ill-defined and its borders are not consistently outlined. For example, the Cad8 (WT) stained region around the tip and continuing along the medial border of V1 is much larger than indicated by the stippled line.*

We thank the reviewers for bringing this important point to our attention. We did not outline the gene expression domains with enough accuracy and consistency in the figures of the original manuscript, and we apologize for that. We have carefully revised the outlines that in the current manuscript version much more accurately delineate the gene expression borders. Moreover we hope that the more detailed definition of HO mentioned above (detailed comments #1) together with the newly added whole mount gene expression images (Figure 3—figure supplement 1–Figure 3—figure supplement 2) that show additional *Cad8*- and *Lmo4*-stained whole brain cases leads to a more conclusive definition of the gene expression borders and how V1 and HO sizes were determined. As mentioned above, we have modified the text accordingly to emphasize this issue (paragraph seven, Results).

*In addition, V1 in ne-Emx2 mice has an unusual shape with an unexpected "tipped nose", indicating that the region enclosed by the stippled line may be larger than V1. The flattening technique can contribute to an overestimation of the size differences reported in the study.*

The comment of the reviewers is correct. In *ne-Emx2* brains the shape of V1 is slightly affected. This is mostly apparent around the anterior tip of V1. The revised outlines now more accurately match the expression domains of the probed genetic markers. We have clarified in the revised manuscript that the measurements in Figure 3 and Figure 4 were made on un-sectioned whole brains and not on flattened cortical sections. Although, performing in situ hybridizations using whole brains is more difficult, compared to in situ hybridizations on sections, we have used the whole brain approach for quantification purposes to prevent potential sizing artifacts due to the flattening and cutting procedures that were mentioned by the reviewers. The images in Figure 3 and Figure 4 show the staining patterns on the intact brain and we therefore remain confident that the quantifications are accurate and reproducible. In the revised manuscript, we more clearly state that whole un-sectioned whole brains were used for the quantifications to prevent confusion of the reader regarding this point (paragraph six, Results).

*7) Figure 4, similar to Figure 3. Cad8 (WT) and Lmo4 (WT) include weakly stained regions in medial extrastriate cortex, which are not included in cKO mice.*

As mentioned in more detail in our reply to the reviewers’ comment #6 above, we again appreciate the reviewers’ attention to important detail and for bringing this important issue to our attention. We apologize for the inaccurate outlines and have revised them carefully.

[Editors' note: further revisions were requested prior to acceptance, as described below.]

Reviewer #1:

*[…] One remaining point: The Abstract needs to be slightly rewritten to bring it into line with the rest of the paper. In the Abstract it is stated: "we have generated bi-directional changes in primary visual cortex size in vivo and have used it as a model to show a novel and prominent function for genetically determined primary visual area size, which also proportionally dictates the sizes of higher order visual areas." It seems (again) that the authors want to say that V1 size controls the size of surrounding higher order areas. Previous review suggested there was not enough evidence for such a conclusion, and pointed to an alternate interpretation: genetic factors intrinsic to cortex control the size of both V1 and HO areas in coordination. The cover letter for the revised MS and the new Discussion section appear to agree with the latter conclusion.*

We agree to the statement of the reviewer. Our data supports the claim that genetic factors intrinsic to the cortex control the size of both V1 and HO areas in coordination. We have modified the Abstract accordingly to bring it in line with the rest of the manuscript.

Reviewer #3:

Although V1 and the organization of surrounding cortex are beautifully illustrated in several newly added images, there are remaining concerns that the revision failed to analyze the outstanding material in a consistent fashion. This impression is based on the inexplicable discrepancy of the definition of HO shown in Figure 1 and Figure 3, which indicates that the authors are unsure how to subdivide the real estate. In Figure 1 it appears that the border of HO runs from the anterior tip of retrosplenial cortex along the posterior border of S1, continues along the border of auditory cortex and ends by intersecting the rhinal fissure. This does not fit the scheme shown in Figure 3, in which HO does not extend to the border of auditory cortex and does not include retrosplenial cortex. If I understand this correctly, HO in Figure 3 is clearly smaller than in Figure 1 and does not include ventral posterior temporal cortex nor does it include retrosplenial cortex. In fact, HO Figure 3 is what was measured in Figure 3 and Figure 4. Maybe these discrepancies were simply an oversight which can be easily corrected. But the lapse heightened my concerns that the size of HO was inconsistently assessed (e.g. in Figure 3 the border of Cdh8-expression in wt is much more medial than indicated and differs from the border in ne-Emx2) and that the substance of the paper is less solid than suggested by the graphs shown in Figure 3 and Figure 4. I recommend re-analyzing the material with a more critical eye on the detailed borders of HO. I am sure that the authors are acutely aware that details matter a lot more in small than large brains.

We would like to mention that we have analyzed the data in a consistent fashion. In general, claims that we have made about the HO as defined in Figure 1 are qualitative (e.g. HO appear enlarged in *ne-Emx2* brains), whereas the measurements of HO in Figure 3 were made without including the retrosplenial cortex.

The reviewer’s concern reflects that our revised manuscript failed to better explain how HO areas reflect slightly different cortical territories in Figure 1, compared to Figure 3 and why this is the case. We have revised the manuscript to better state this discrepancy and why and have added some new data that better explains why we exclude retrosplenial cortex from our quantifications in Figure 3 and Figure 4 (details are explained below).

In Figure 1 we have used 5HT staining to introduce the central issue of the paper: Upon genetically increasing V1 size, the surrounding higher order areas appear to be qualitatively enlarged as well. We have chosen to use 5HT staining to introduce this central issue in order to present the analysis in a way that is comparable to virtually all studies that have analyzed and demonstrated primary area patterning by using 5HT staining as a ‘gold standard’ of the cortical areal layout and to extend on them in a consistent fashion by highlighting altered higher order areas by using the same technique. Our main focus is using V1 as a model to probe the influence of altered V1 size on the proportions of higher order visual areas. An issue that is not understood. As mentioned in our initial review comments, the caveat is that 5HT staining does not reveal the retrosplenial cortex and hence the HO complex that we needed to define initially in Figure 1 to set the stage for the rest of the data includes it. This important consideration is clearly stated to the potential readership of the manuscript in the definition that can be found in paragraph one, Results.

The dilemma, which might lie on the bottom of some of the reviewer’s concern, might be that especially in the mouse, there is no commonly agreed upon or long-established terminology for the cortical region of interest. At least this is the case for the rodent brain. Descriptions in published studies are in itself inconsistent ranging from ‘extrastriate cortex’ to ‘higher order visual areas’ and ‘higher order cortex’. All of these previously used terms are slightly inaccurate. For example, there is no such thing as a striate cortex in rodents, but it is common to describe higher order visual areas in rodents as such since there appears to be a general agreement that extrastriate areas in rodents reflect higher order areas around V1 and that these areas are analogous to the striated/extrastriated areas that are characteristic for higher, more complex mammalian brains like cats, monkeys, and humans.

In the initial manuscript version we therefore referred to this cortical territory as higher order visual areas (VHO), which was felt to be incorrect. We agreed upon this view and that it would be more appropriate to describe this territory more broadly as higher order area complex (HO) throughout and define better which cortical subdivisions are included and excluded in Figure 1 versus Figure 3 and Figure 4.

Compared to 5HT staining in Figure 1, we in Figure 3 use genetic markers that enabled us to distinguish between higher order visual areas medially of V1 and the retrosplenial cortex around the cortical midline. In this quantitative assessment, we have therefore excluded the retrosplenial cortex from our definition of HO since genetic markers enabled us to make more precise anatomical claims about the core areas of interest (mainly higher order visual areas) that were assessed quantitatively in Figure 3 and Figure 4. The main focus of our study is genetic scaling between primary visual cortex and higher order visual areas. Also, due the fact that the retrosplenial cortex is not a higher order cortical area (Garrett et al. 2014, Kalatsky and Stryker 2003, Marshel et al. 2011, Vann, Aggleton, and Maguire 2009) and due to unpublished data from our lab (see below for details), we had reasons to exclude the retrosplenial cortex from the quantifications of HO in this study. Overall we do believe that the claims of the manuscript are solid and valid, although HO in Figure 3 and Figure 4 reflect a smaller cortical territory, compared to Figure 1, because the anatomical cortical subdivisions are accurately described in each case. We have nonetheless made the attempt to improve the clarity of the text and have revised the manuscript to more explicitly state that the definitions of HO in Figure 1 and Figure 3 and Figure 4 are slightly different (paragraphs one, seven and eight, Results).

We also have decided to include an additional piece of unpublished data into the manuscript that elaborates specifically on the scaling of the retrosplenial cortex and how it differs from higher order visual areas as measured by *Lmo4* and *Cdh8* staining: To complement the *Lypd1* WMISH staining in wt brains, we have added *Lypd1* WMISH staining on *ne-Emx2* brains into Figure 3—figure supplement 1 and have quantified it. The *Lypd1* gene expression domain around the territory of the retrosplenial cortex is significantly larger in *ne-Emx2* brains (114.3% ± 5.2%, p = 0.0225, n = 4), compared to wt brains. But importantly, this size increase is NOT proportionate to the size increases of V1 (~145%) and HO as measured by *Lmo4* and *Cdh8* (~145%) staining reported in Figure 3. This suggests that increased V1 size is specifically accompanied by a proportionate size increase of related higher order visual areas. This new data explains why we have avoided including the retrosplenial cortex area into the quantitative assessment of higher order visual areas. This new data also helps to more precisely define the central finding of the manuscript: In the revised manuscript we now state that genetic mechanisms control the linear scaling between RELATED cortical primary and higher order sensory areas (e.g. vision). To incorporate these changes and to bring the manuscript in line with the presented data, we have modified the title slightly, have added a passage describing the new data (paragraph seven, Results) and refer to *Cdh8/Lmo4* labeled HO as ‘related HO’ or ‘higher order visual areas’ throughout the main text and Discussion.

The schematic in Figure 3 was intended to serve as a visual aide to show location and approximate perimeters of primary and higher order visual areas as defined previously (Wang and Burkhalter 2007) and their overall similarity to the expression domains of *Cdh8* and *Lmo4* surrounding V1. We have removed the schematic from the revised Figure 3.

[Editors' note: further revisions were requested prior to acceptance, as described below.]

Reviewer #3:

*1) To resolve the conflict between the measured HO (Figure 3, Figure 4) and HO defined by 5HT staining (or the virtual absence of 5HT expression) in Figure 1 suggest renaming it HO-5HT to indicate the conditions under which the observation was made and unequivocally state that HO and HO-5HT are not the same.*

We thank the reviewer for the constructive comment and suggestion to resolve potential misunderstandings about slightly different definitions of higher order areas derived from stainings using different sets of markers. We have followed the suggestion and have termed the HO as stained by 5HT as HO-5HT throughout the text, in Figure 1 and its figure legend. Consistently, we have also amended the annotation of HO as labeled by *Rorb* in Figure 1 as HO-*Rorb* in Figure 1 and its legend.

*2) HO includes at a minimum visually dominated as well as higher order auditory areas. I therefore recommend revising the sentence (Abstract) to: "… surrounding higher order sensory areas". I further recommend that similar revisions be made throughout the text to make it absolutely clear that the cortex measured here is more extensive than extrastriate visual cortex.*

We thank the reviewer for the recommendation. We would like to clarify that the complex of higher order areas that was quantified using *Lmo4* and *Cdh8* marker genes does not include primary auditory cortex or secondary auditory areas. As it is clearly shown in Figure 3—figure supplement 1 A and B (as an example using *Lmo4* staining), primary and secondary auditory areas are included in the complex of auditory areas that we have termed “Aud”. As it is evident in three different angles in Figure 3—figure supplement 1, this “Aud” complex is excluded from our measurements. We have currently another paper in review that in high detail is concerned about the scaling of the auditory areas in comprehensive sets of different inbred mouse strains and transcription factor mutants. Unfortunately it is too premature to reference at this point, but we would like to emphasize that we have solid evidence about how primary and secondary auditory areas relate to expression sub-domains of *Lmo4* and other area markers that are expressed in this cortical region in addition to the well documented and clearly stated exclusion of secondary auditory areas (defined as ‘Aud’, e.g. Figure 3—figure supplement 1) that are already present in the manuscript. Considering these points we would like to re-emphasize that secondary auditory areas are excluded from the presented measurements and that we remain confident that our claims (e.g. Abstract) are in line with the presented data.

*3) I recommend revising the border of HO (wt/Cdh8) in Figure 3. It is clearly evident in the image that the Cdh8 expression includes a wedge medial to V1 which extends to the tip of the arrow head. A similar adjustment of the medial border is necessary for ne-Emx2 Cdh8 (Figure 3). The same revisions are necessary in Figure 4 in which the medial borders of Cdh8 expression in wt and cKO are more medial than indicated.*

We would like to acknowledge that the reviewer has an exceptional eye and commitment to detail. We agree that around the area in question (around arrowheads in Figure 3), there is some *Cdh8* staining detectable. However, we would like to mention that this gene expression domain shows much lower staining compared to the outlined complex of all the other regions around V1 that we have quantified (please re-examine *Cdh8* whole brain images in Figure 3—figure supplement 2). We have mentioned this issue consistently in all previous versions of the manuscript as a discernable difference between the expression domains of *Lmo4* and *Cdh8* (arrows in Figure 3, see paragraph seven, Results): Due to much lower expression compared to all the other more lateral regions around V1 (e.g. Figure 3—figure supplement 2), we have excluded this domain from the measurements and consistently from the outlines. Due to these criteria we think that the present outlines are well defined on the basis of similarly high *Cdh8* gene expression levels, compared to detectable but much lower levels in the region of question. We feel that modifying the outlines to include this area of lower *Cdh8* expression could potentially appear inconsistent to potential readers and would also require us to comment on this issue and criteria (that compared to as is, would not be based on similarly high *Cdh8* expression levels) in much more detail in the text. Since this issue has always been well documented (arrowheads in Figure 3) and mentioned in the text (paragraph seven, Results) by comparing *Lmo4* and *Cdh8* expression levels in this region, we have the impression that altering the outlines would not be beneficial. To the contrary, it would require us to re-define gene expressions areas without adding additional insights to the conclusions of the manuscript. We have revised the text slightly for clarity to emphasize more progressively that the excluded region does in fact expresses *Cdh8*, but at much lower levels, compare to the outlined and defined area complex of consistently much higher *Cdh8* gene expression (paragraph seven, Results).